# Revenge in Couple Relationships and Their Relation to the Dark Triad

**DOI:** 10.3390/ijerph18147653

**Published:** 2021-07-19

**Authors:** Miguel Clemente, Pablo Espinosa

**Affiliations:** Department of Psychology, Research Group in Criminology, Legal Psychology and Penal Justice, Universidade da Coruna, 15071 A Coruna, Spain; pablo.espinosa.breen@udc.es

**Keywords:** psychopathy, Machiavellianism, dark personality, romantic relationships, retaliation

## Abstract

Background: This research examines how, when a romantic partner commits a perceived transgression that leads to couple break up, vengeful reactions are predicted by the type of transgression and the Dark Triad of personality. Methods: An incidental sample of 2142 participants, half male and half female aged 18 to 70, completed a questionnaire developed by the authors to assess how they had reacted after being the perceived victims of a transgression committed by their partner and a measure of the Dark Triad. Results: Results show half of the people who feel as though they are victims of a partner transgression show revenge reactions. These reactions are more emotional than rational and do not usually anticipate their consequences or success. Moreover, revenge is related primarily to psychopathy and to a lesser extent to Machiavellianism. Psychopathy is the best predictor for revenge thoughts and actions, whereas narcissism does not predict revenge when controlling for other dark traits. Conclusions: This study contributes to the explanation of revenge reactions in couple relationships in relation to the type of transgression perceived and the Dark Triad. Conflicts that arise out of revenge may have long-lasting consequences for both the perceived aggressor and victim, and our results may be useful for assessing risks, monitoring, and preventing negative consequences for partners or ex-partners.

## 1. Introduction

This research examines whether the thoughts and actions of people who take revenge in retaliation after suffering some perceived harm are related to Dark Triad traits: Machiavellianism, subclinical psychopathy, and subclinical narcissism [1]. Revenge is a motivation for crime and detrimental behavior, and it has been present in the literature of all ages, sometimes describing it, other times encouraging its fulfillment. Among the detractors of revenge, everyone is familiar with the sentence of Confucius: “Before you embark on a journey of revenge, dig two graves”. At the opposite end is the well-known novel “The Count of Monte Cristo”, by Alexander Dumas, which made famous the idea that revenge is best served cold and can take years to be ready. Cinema has been prolific in dealing with the issue of revenge, and Quentin Tarantino’s *Kill Bill* is a good example of it. We prefer the phrase of Albert Einstein: “Weak people revenge. Strong people forgive. Intelligent people ignore.”

It is difficult to calculate the number of crimes committed in revenge, especially as most statistics do not include criminal motivation. Nevertheless, most homicides are motivated by revenge [2]. Revenge is an ever-present, long-lasting phenomenon that may be construed as a health problem considering its consequences for the victim and the perpetrator, and it sometimes is executed years after the event that triggered it has finished. Revenge is prevalent in couple relationships, and over 90% of people in a couple relationship report “getting even” in the past with their partners [3].

Revenge has been compared to aggression [4], and it is construed as a behavioral manifestation. Revenge may be individual or group, and individual revenge usually manifests more intensely than group revenge [5]. Prior to revenge, there must be a perception of having suffered an injustice, that this injustice was carried out by specific individuals who they consider guilty, and that there must be a violent retaliation against that individual or group blamed for the perceived aggression [4]. Therefore, violence is directed against the person that the alleged victim identifies as causing the harm suffered, and intentionality is attributed to the perceived aggressor [6,7]. Thus, the perceived victim becomes an aggressor, often without evidence of an actual wrongdoing from the recipient of revenge. Often, the aggressor justifies their actions, claiming that they are the actual victims [8].

The perception of being unfairly treated leads to anger, which is an emotion associated to revenge [9,10]. The relationship between anger and revenge has inspired research that examines these two variables in relation to Dark Personality. Previous research has found that vengeance is correlated to the Dark Triad [11], and evidence of a specific relationship between narcissism and revenge has also been found [12,13,14]. Further research [15] found that Machiavellianism and psychopathy are related to emotional vengefulness and negatively related to justice decisions. More specifically related to the present study, the Dark Triad predicts revenge against a romantic partner after an infidelity, as they are related to an increase in the perception of revenge effectiveness and the endorsement of power and justice goals. [16]. In cases of infidelity, the Dark Triad, and especially psychopathy, predicts every type of revenge except ending the relationship [17]. The Dark Triad also predicts jealousy motivated revenge [18].

Revenge in couples usually occurs when the relationship ends and one of the partners does not forgive the other for some perceived humiliation, such as an infidelity or break up [19,20,21,22]. Revenge has been conceived as an alternative mechanism to the justice system that often involves breaking the law. It may happen when justice “subjectively” fails to restitute the alleged victim [4]. However, in cases of conflict between parents, this resource is used even when the justice system provides a solution [23,24]. Perhaps one of the characteristics of revenge is that it takes place despite the justice system’s actions. This may happen because the perpetrator considers that the perceived harm must be directly repaired beyond the solutions provided by law and society. This leads to the notion of revenge as a harmful behavior aimed at inflicting pain to others, and it is accompanied by resentment, which is a concept that is excluded from penal codes but not from personal ones. Revenge, especially when it involves inflicting harm, is negatively related to forgiveness, which in turn is associated to higher positivity and happiness [25,26]. The usefulness of revenge and what the avenger achieves has been much debated [27,28]. However, research finds no evidence of positive personal outcomes resulting from revenge [4,29].

Revenge is more intense when it involves a romantic partner [30,31]. Moreover, when revenge is carried out by people who were emotionally close, perpetrators use all kinds of elements within reach, such as gossip, hurtful remarks to acquaintances, coercive actions, harassment, etc. [32,33]. In this context, the aggressor seizes the concept of "virtuous violence" [34], so that revenge is justified as the only option left to deal with their perceived offense. The perpetrator seeks support from people in their environment, and revenge becomes a social and moral obligation, even endorsed by social values sometimes biased against women, as it happens in societies with a greater culture of honor [28,35,36,37]. Another defining characteristic of revenge is that it does not happen when the target of revenge is perceived as more powerful than the perpetrator, and in fact perpetrators usually perceive they have both power and right to exert revenge [19,35,38,39].

This study aim is to examine whether, in a couple conflict, the type of perceived transgressions by the partner is related to revenge decision making. Additionally, this study aims to determine to what extent the Dark Triad influences revenge when a member of the couple feels aggrieved and decides to act against the perceived aggressor.

It is hypothesized that both emotional and cognitive aspects are involved in the process of revenge. Hence, our first hypothesis is that revenge will be positively associated with feelings of relief and satisfaction. Conversely, our second hypothesis is that revenge will be negatively related to planning and anticipating outcomes. The third hypothesis in this study is that the Dark Triad of personality will be positively related to revenge. According to the literature [40], this can occur through different processes. Subclinical psychopathy is the variable most strongly associated with behaviors harmful against others because it involves callousness and insensitivity. In turn, damage to self-esteem and the ego threat of receiving an offense can make narcissistic individuals eager to take revenge to compensate for their feelings of insecurity. Machiavellianism, besides, is associated with manipulative behaviors with a higher level of planning to obtain tangible results, which matches the anticipatory behaviors associated to revenge. In any case, the three traits contribute to a common core related to harmful behavior toward others because they all have common elements such as selfishness and lack of empathy [41].

## 2. Materials and Methods

### 2.1. Participants

A series of criteria were established for sample selection: Participants were chosen among people who had had a couple relationship, which had lasted at least six months, and that resulted in a breakup due to a negative interaction event experienced as detrimental that the respondent blamed on the other partner and led them to consider revenge (whether they had carried it out or not). In addition, sampling was stratified: participants had to be adults (over 18), 50% female and 50% male, belonging to four age groups, in equal proportions (18 to 25 years, 26 to 35, 36 to 50, and more than 51). Undergraduate students were given the option to collaborate in the study in exchange for course credit. These collaborators were briefed about the eligibility criteria and asked to contact suitable participants within their social networks, and then, they provided participants with a link to complete an online questionnaire. Thus, the sample was incidental and additionally, a snowball procedure was used, so participants who met the criteria were asked to provide contact with other eligible participants. In this way, 3942 participants were contacted, among whom there were 2744 who met the criteria (having had a partner and having had some conflict with them). A series of control questions were used to detect participants answering the items without paying attention, following procedures to screen careless responding [42]. Hence, 194 participants had to be eliminated, as there were suspicions that they had not paid sufficient attention when answering, so the final sample included 2550 participants.

### 2.2. Instruments

#### 2.2.1. Revenge Survey

We developed a survey about the revenge-related reactions provoked by a conflict within a couple relationship. Participants who indicated a conflict with a romantic partner were asked to describe it and answer a series of “yes” or “no” questions related to revenge. These questions were based on forensic professional practice and were grouped as follows:

Perception of being hurt: Revenge involves the perception of being the victim of an intentional harm. Operationally, this aspect was measured by the following questions: “Do you think that person harmed you on purpose?” and “Do you think they wanted to hurt you?”

Evaluation of alternatives by the alleged victim when feeling hurt: People take revenge when dialogue is not possible, resorting to justice is unthinkable or unusable, or when they are not willing to let the conflict go without revenge. The following questions were asked to measure these notions: “When that happened to you, did you talk to your partner or ex-partner to try to solve it?” “Could you solve it?” “Did the problem persist after the dialogue or was it solved?” “Did you turn to the police or the justice system to solve the problem?” “Could you solve it?” “Did the problem persist after going to the police or the justice system or was it solved?” “Did you decide to assume it was better to do nothing and forget the offense?” “Could you solve it like that?” “Did the problem persist after your intention to do nothing against your alleged aggressor or was it solved?”

Evaluation of the chances of successful revenge: When someone thinks revenge is the best possible solution, they consider whether revenge can succeed. Operationally, the following questions were asked, focusing on the decision to take revenge: “When you decided to act against the other person, did you appraise whether you could succeed in doing so?” “Did you determine how to act so that the other person would be harmed but you would be safe from possible attacks?” “Did you consider whether attacking the other member by force or contacts or economics would be effective and feasible?”

Finally, we wanted to know whether the individual expected to feel better emotionally after revenge. For this purpose, one question was asked: “Did you think that attacking the other person would relieve you emotionally and reassure you?”

#### 2.2.2. Dark Triad Questionnaire

Participants also responded to the Short Dark Triad (SD-3) scale [40]. This instrument consists of 27 items, 9 for each trait in the Dark Triad. For this study, we used the Spanish version [43]. The response format is a five-point Likert type scale (Totally disagree (1), Disagree (2), Neutral (3), Agree (4), and Totally Agree (5)). An example of an item (measuring Machiavellianism) is: “It isn’t smart to tell your secrets”. Reliability for this scale in our sample was: Machiavellianism, 0.792; Subclinical Narcissism, 0.711; Subclinical Psychopathy, 0.777. Therefore, reliability rates were considered appropriate.

### 2.3. Procedure

Participants read a brief description of the study and gave their informed consent. Some key questions that were intended to verify that participants had read the questions and responded appropriately (for example, “This question is to verify that you are attentive to each issue. Please indicate alternative 1”). Participants were required to answer every item, and the data collection procedure did not allow missing responses.

Participants’ responses to the reported conflict were classified into four broad categories described in the results.

Prior to performing the investigation, permission was requested from the Research Center Ethics Committee of the corresponding author. The research meets the ethical criteria of the Helsinki protocol and the American Psychological Association.

### 2.4. Statistical Analysis

After categorizing the descriptions of conflicts given by participants, we examined differences in the revenge survey between the resulting types of conflict using χ^2^ tests. In addition to this descriptive analysis of the sample, *t*-tests were performed to determine whether there were significant differences in participants’ score in the Dark Triad variables depending on their responses to the revenge survey. Logistic regression analyses were also carried out using as criteria of participants’ responses to revenge-related behaviors and thoughts and using as predictors the variables of the Dark Triad, the participants’ sex, and the type of conflict suffered. In this way, the role of each predictor could be verified, controlling for the effect of the rest of the predictors on the model. Data analyses were carried out with IBM SPSS 27.0 (Armonk, NY, USA).

## 3. Results

Participants’ responses about the type of conflict experienced with their partner were grouped into four categories. These qualitative categories were conceptually developed by the authors based on forensic practice and were very broad to include most conflicts described by the participants. Categories considered that conflicts were either related to communication, trust, sex, or aggression, and a particular conflict description could fit more than one category (e.g., “My partner felt jealous and used to beat me” would relate to both trust and aggression). The advantage of using very broad categories was that responses were easier to identify as belonging to a category and that no category had a small number of cases compared to others, although about one in 20 responses did not fit in any category. A sample of 100 descriptions was categorized separately by two judges, and they fully agreed.

The most frequent category referred to communication problems, including arguments, disrespect, distancing, and isolation from the partner (33.0% of men and 30.0% of women). The other categories related to sexual infidelities committed by the partner (30.8% of men and 22.8% of women; jealousy, control, and harassment by the partner (17.9% of men and 17.0% of women); and aggressions, either physical, verbal, or psychological (11.7% of men and 24.7% of women). In addition, 6.7% of men and 5.4% of women indicated conflicts that did not correspond to any of these four categories (e.g., theft or drug use).

The percentage of responses to questions about revenge and χ^2^ scores were calculated to determine whether the differences between the responses were significant depending on the type of conflict. The results are presented in Table 1.

The thought of revenge (information collected before discarding the participants who did not consider revenge) happens in 43.4% to 61.2% of cases. That is, even though the participant perceived that their partner or ex-partner meant to hurt them, only half of them thought about revenge. Revenge happened to a greater extent in infidelities and aggressions.

Among those who thought about revenge, between 8.6% and 22.2% them thought about attacking people close to their partner in this process of revenge, particularly in cases of infidelity.

Between 34.3% and 59.3% of the participants perceived that those who hurt them did so intentionally. This perception is higher in the case of being the victim of aggression. We also examined whether thoughts of vengeance depended on the attribution of intentionality, and results showed a significant relationship (χ^2^(1) = 139.969, *p* < 0.001). Thus, 66% of the participants who perceived intentionality but also 42% of those who did not perceive such intentionality thought about revenge.

The vast majority acknowledged that they used dialogue to try to restore normality in the relationship (90.1–75.3%), but only in one-third of the cases (36%) did dialogue achieve adequate results, although this is a significant improvement compared to the 18% who reached a solution without talking with their partner (χ^2^(1) = 53.133, *p* < 0.001). Nearly half of those who felt hurt reported that the problem subsequently continued. Infidelity is the problem with the least tendency to dialogue.

Having to go to the police or the justice system was infrequent except in the case of aggression, where 21.7% of participants report they would go the police.

As to whether acts of revenge are carried out mainly rationally or emotionally, between 29.3% and 47.4% of the people who use revenge ponder whether their actions can succeed when they perform them, and between 20.5% and 35.2% consider whether their acts of revenge risk being discovered. In both cases, issues of infidelity stand out.

Finally, performing revenge did not always imply the anticipation of emotional relief, as this relief only occurs in 33.1% to 50.1% of the cases. Relief after revenge stands out in infidelity conflicts.

Next, *t*-tests were performed to examine differences in participants’ scores in the Dark Triad variables depending on their responses to the revenge survey. Table 2 below summarizes these results.

Subsequently, a logistic regression analysis was carried out using as a criterion the different revenge-related thoughts and behaviors and as predictors the variables of the Dark Triad, as well as sex and the type of conflict. As for the categorical predictors in the model, in the case of sex, men were taken as a reference category to make comparisons. For the type of conflict, the reference category was the conflicts not classified within any of the four defined categories. As a contrast method, we used the difference of each type of conflict with the mean of the other categories. Table 3 shows the results and the significant predictors for each thought and behavior related to revenge.

The model for thoughts related to revenge on the partner provided a correct prognosis of 62.8% of the cases, compared to 51.8% if all cases were assigned to the largest category. As for the effect of each predictor on the model, keeping constant the other predictors, Machiavellianism and psychopathy were significant predictors of thoughts of revenge, but narcissism was not. Probability ratios indicated that for each increase of one point (on a scale of 1 to 5) in the Machiavellianism score, the odd ratio (OR) showed that the odds of having vengeful thoughts were multiplied by 1.4. For psychopathy, this increase was almost 1.7. Sex was found to be a significant predictor of thoughts of vengeance. The estimated probability ratios indicate that women are 1.4 times more likely to have thoughts of revenge than men. All conflicts were significant predictors of revenge, but not in the same direction. Whereas infidelity and aggression increased thoughts of revenge, suffering jealousy and control and communication conflicts were associated with a lower probability of thinking about revenge (1.5 times less likely in jealousy and control and 1.4 times less for communication problems).

The model based on thoughts related to taking revenge against third parties was significant, although it did not improve the prognosis compared to the initial 85.1%. The only variable of the Dark Triad that predicted revenge against third parties was psychopathy, and each point of increase in the psychopathy score multiplied by 2.5 the odds for revenge against third parties. Sex was not a significant predictor in this model, whereas suffering aggression and suffering jealousy and control predicted a decrease to about one-half in the chances of having these thoughts compared to other types of conflicts.

As for the partner’s attribution of intentionality to harm, the model provided a correct prediction of 64.8% of the cases, which was a modest improvement over the initial 59.6%. Machiavellianism and psychopathy predicted an increase in the likelihood of attribution of harmful intentionality, whereas narcissism, controlling for the effect of other variables, predicted a decrease. Sex did not predict attribution of harmful intentionality but suffering aggressions did predict attribution, 2.6 times more often than in other conflicts. In contrast, jealousy, and control and communication problems were associated with a lower perception of harmful intentionality.

Regarding attempts at dialogue after being harmed by the partner, the model indicated a good fit but did not improve the initial prognosis of 84.6%. However, it indicated that psychopathy reduced the chances of dialogue, and for each additional point in psychopathy dialogue, it became 1.58 times less likely. The other variable that reduced the likelihood of dialogue were conflicts over infidelity. Compared with other conflicts, dialogue was 1.8 times less likely in the event of infidelity. There was no significant effect on aggression, but jealousy and control and communication problems were associated with a higher likelihood of dialogue. As far as finding a solution, no variable of the Dark Triad or of sex were significant predictors. The results only showed that conflicts related to infidelity were 2.2 times less likely to be solved, and those related to communication were 1.4 times more likely to be solved than other conflicts.

The predictive model on the answers about whether the problem continued after seeking a solution showed a modest improvement in the prognosis from the initial 50.2% to 55.8%. Psychopathy increased the chances of the conflict reappearing. Compared to men, women perceived to a greater extent that the conflict had not been definitively resolved. The results also indicate that conflicts related to aggression and jealousy and control are the ones that last longer.

Regarding resorting to the police or the justice system in response to the conflict, which was indicated by 8.7% of the participants, the model did not improve the base prediction of 91.3%. The only variable of the Dark Triad that was a significant predictor of going to the police or justice is Machiavellianism. Compared to men, women reported more often that they would go to the police or justice. There were also differences according to the type of conflict. Both infidelities and communication and distancing problems were associated with decrement in police or judicial complaints, reducing the odds to one-half or one-third. Not surprisingly, aggressions were clearly associated with a greater tendency to report, multiplying by 3.5 the probabilities with respect to other conflicts.

The model for participants’ appraisal of success when considering revenge went from a base prediction of 63.4% to 68.9%. Machiavellianism and psychopathy but not narcissism predicted a positive response to the assessment of success. Compared to men, women appraised the success of revenge more. Differences were also found according to the type of conflict. Both infidelities as well as jealousy and control significantly predicted the assessment of success but in opposite directions.

In terms of assessing not being discovered, the model modestly improved the base prediction from 73.7% to 76.4%. Machiavellianism and psychopathy predicted a positive response in the assessment of not being discovered. Again, narcissism was not a significant predictor when controlling for the effect of other predictors. Women value protecting themselves against reprisals more than men, and there were also differences according to the type of conflict for conflicts caused by jealousy and control and communication problems, in the sense that there was a decrease in valuing protecting oneself against retaliation.

Finally, the predictive model for emotional relief achieved by revenge was examined. The model showed an increase in prediction from 60.7% to 68.1%. The model was significant but did not obtain a good fit. Machiavellianism and psychopathy predicted a positive response on emotional relief, but narcissism did not. Women were more likely to experience emotional relief than men. Differences were also found according to the type of conflict for conflicts caused by infidelities, which doubled the chances of experiencing emotional relief, and jealousy and control, which reduced the probabilities of feeling relief after revenge.

## 4. Discussion

This research has examined the predictive capacity of the dark personality traits in the process of revenge for an offense committed by the partner. Regarding our first hypothesis about emotional aspects involved in this process, results showed that revenge is associated with emotional relief or satisfaction and is predominant in cases of infidelity. We also found support for our second hypothesis: cognitive aspects of appraisal of success and avoiding being discovered are perceived as less important than emotional aspects and are related mainly to the dark traits of Machiavellianism and psychopathy. Regarding our third hypothesis, Dark Triad traits are consistent predictors of revenge and its planning, especially psychopathy and Machiavellianism, although narcissism is not a significant predictor, which goes against our expectations. There are significant differences in narcissism in the *t*-tests, but when all the predictors are introduced into a logistic regression, narcissism loses predictive power. This effect is probably because narcissism shares variance with Machiavellianism and psychopathy as there is a common explanatory core to all dark traits and in addition, narcissism is the “mildest” of all the dark personality traits [41]. However, both Machiavellianism and psychopathy maintain their predictive power as predictors of revenge when controlling for the other variables in the model. Thus, revenge is associated both with cognitive and emotional components, and this is especially relevant in the case of infidelities, since ORs show that appraisal of revenge success and emotional relief both increase in this type of conflict. Apart from the reported role of Machiavellianism and psychopathy, other dark traits might be specifically related to cognitive and emotional components of revenge. Within the planning components, more efforts are made to harm than to protect oneself against possible retaliation, suggesting that the dark trait of malevolence [41], which entails seeking to harm to others even if one suffers harm, may be involved in the process.

An increase of revenge-related thoughts and behaviors related to the type of conflict was also found. Infidelity and aggression are the conflicts that provoke greater revenge-related reactions. Sex is also a relevant variable to explain the revenge process, as women contemplate revenge more, show greater planning, and experience greater emotional relief and satisfaction. In any case, it should be stated that this effect was obtained keeping the values of the Dark Triad variables constant when comparing males and females. However, men generally score higher than women in these variables [39], which are associated with revenge, so this may be mitigating gender differences. To investigate this effect further, possible differences between men and women should also be examined in acknowledging harmful behaviors toward their partner.

Other results show distinct characteristics of revenge. Around half of the participants who feel aggrieved by the other partner think about revenge, while the rest prefer not to act against their partner or ex-partner. When revenge takes place, it sometimes targets not only their partner but sometimes includes other family members and relatives. For retaliation to take place, it is not necessary for the alleged victim to perceive intentionality in their partner, because in about half of the cases in which no harmful intention is attributed, the participant still thinks about revenge. Dialogue to try to re-establish the relationship or avoid harm occurs on a regular basis, but its results are usually not positive. In addition, especially with people who have high levels of subclinical psychopathy or in cases of infidelity, dialogue is less likely to be considered. Resorting to the police or the justice system is uncommon, although it stands out in cases of aggression. Certainly, in other conflicts, there is no criminal offense, so the role of the judicial system is not relevant.

This research has several limitations, which should be corrected in future research. First, concerning the fact that the sample was obtained incidentally, it would be more appropriate for future studies use random samples. Another limitation that should be addressed in future research is that we did not include other measures of revenge to assess the validity of our research questionnaire. Types of conflict categories were developed by the authors, which is a limitation since they lack sufficient theoretical back up and need to be further tested in other samples to fill this research gap. The categories used in this study may be not precise enough or other types of conflict (i.e., economic disputes) may be relevant in different settings. It would also be adequate to measure sexism [44,45] in future studies. This is relevant because our results show that up to 60% of people consider getting even when they feel wronged by their partner, so conflicts that lead to revenge appear to be closely related to intimate partner aggression. While this would mean that data collection would be more time consuming for participants, sexism would be very relevant for the explanation revenge as it relates to IPA [46,47]. Finally, we believe that future research should also add other personality variables to provide a comprehensive explanatory model.

This research has several implications for practitioners. Feeling aggrieved and seeking revenge in a couple conflict is greatly influenced by the perception of being victimized and attributing most or all responsibility for the conflict to the partner, which is seen as an aggressor. For instance, within the same conflict, a partner may feel as a victim of infidelity, while the other partner may feel he or she is a victim of jealousy, and both may take revenge on each other. Practitioners must take into account how these perceptions of being victimized influence revenge and promote reappraisals to modify biased perceptions. Our results also show that some types of conflicts are more related to revenge. Thus, interventions ought to consider the type of couple conflict as a risk factor for retaliation. Another risk factor that should be monitored for is the dark personality traits of the partners in conflict, which are relevant predictors of revenge. Both of these variables are useful in profiling couple conflicts with a high risk of provoking revenge behaviors.

Lastly, this work deals with a problem that, given the high number of couples affected, has become a public health problem. On another hand, we must bear in mind that most crimes, and especially abuses against partners and ex-partners, are inspired by revenge [3]. We also believe that the issue of the relationship between revenge and forgiveness [25,26], as a way of overcoming and avoiding such aggressive behavior, should be examined in future research. In any case, we believe that these data can help to explain a phenomenon that society is required to solve.

## 5. Conclusions

Revenge after a perceived transgression within a couple relationship that leads to couple break up is predicted by the type of transgression and the Dark Triad of personality. ORs show that infidelity is the type of conflict that is more related to indicators of revenge (thinking about revenge, appraising success of revenge, and feeling relieved after exacting it). Conversely conflicts related to jealousy and control trigger less revenge responses and elicit more dialogue, perhaps because individuals subjected to this conflict just want to end the control exerted by their partners. Possibly, in this type of conflict, individuals feel less power to exact revenge [35] since their partners are in a more dominant role.

Sex is also a predictor of most reactions related to revenge, with females scoring higher than males, and future research must examine the role of gender in revenge. Finally, Dark Triad traits are significant predictors of revenge thoughts, emotions, and actions. Psychopathy is the strongest predictor, followed by Machiavellianism, while Narcissism effect disappears when controlling for Psychopathy and Machiavellianism.

All these results are useful in assessing situations where the type of conflict between the couple and personal characteristics related to dark traits may pose a risk for the well-being of the partners or ex-partners.

## Figures and Tables

**Table 1 ijerph-18-07653-t001:** Response percentages for each reaction to perceived transgressions committed by the romantic partner.

Action	Infidelity	Aggression	Jealousy, Control	Communication	χ^2^
Thinking about revenge	61.2%	60.8%	45.7%	43.4%	70.939 *
Thinking about harming others	22.2%	12.9%	8.6%	12.6%	50.257 *
Attributing intentionality	39.9%	59.3%	34.5%	34.3%	91.813 *
Dialogue to solve it	75.3%	84.4%	90.1%	89.4%	69.262 *
Solution after dialogue	24.8%	32.1%	30.9%	40.0%	45.012 *
Problem continues after dialogue	53.5%	43.0%	43.7%	53.1%	23.730 *
Resorting to police or justice	4.4%	21.7%	7.7%	4.7%	134.213 *
Appraisal of revenge success	47.4%	34.6%	29.3%	33.7%	49.586 *
Valuing not being discovered	35.2%	27.6%	20.5%	21.2%	46.185 *
Emotional relief of revenge	50.1%	42.8%	33.1%	33.1%	59.796 *

* *p* < 0.001.

**Table 2 ijerph-18-07653-t002:** Averages and standard deviations of the variables of the Dark Triad as a function of vengeful actions.

Action		MACH	*t*	NA	*t*	PP	*t*
Thinking about revenge	No	2.67 (0.76)	9.95 *	2.52 (0.63)	5.59 *	1.92 (0.66)	11.03 *
	Yes	2.98 (0.82)		2.67 (0.75)		2.25 (0.81)	
Thinking about harming others	No	2.78 (0.79)	7.95 *	2.55 (0.68)	8.22 *	2.01 (0.72)	13.96 *
	Yes	3.13 (0.85)		2.87 (0.76)		2.58 (0.81)	
Attributing intentionality	No	2.73 (0.77)	7.27 *	2.57 (0.65)	3.26 *	1.99 (0.69)	8.16 *
	Yes	2.97 (0.84)		2.66 (0.77)		2.24 (0.82)	
Dialogue to solve it	No	2.98 (0.80)	4.01 *	2.72 (0.71)	3.48 *	2.33 (0.86)	6.78 *
	Yes	2.80 (0.81)		2.58 (0.70)		2.05 (0.73)	
Solution after dialogue	No	2.83 (0.82)	0.14	2.60 (0.74)	0.49	2.11 (0.77)	1.41
	Yes	2.84 (0.78)		2.61 (0.63)		2.06 (0.73)	
Problem continues after dialogue	No	2.79 (0.79)	2.55 ^+^	2.58 (0.67)	1.74	2.04 (0.74)	3.06 ^+^
	Yes	2.87 (0.83)		2.63 (0.73)		2.14 (0.77)	
Resorting to police or justice	No	2.81 (0.81)	3.69 *	2.60 (0.70)	0.59	2.08 (0.76)	1.76
	Yes	3.02 (0.83)		2.63 (0.70)		2.18 (0.77)	
Appraisal of revenge success	No	2.67 (0.76)	13.58 *	2.50 (0.64)	10.36 *	1.92 (0.68)	15.92 *
	Yes	3.11 (0.82)		2.79 (0.76)		2.39 (0.80)	
Valuing not being discovered	No	2.70 (0.76)	14.35 *	2.52 (0.65)	10.72 *	1.96 (0.69)	15.91 *
	Yes	3.20 (0.82)		2.85 (0.79)		2.48 (0.81)	
Emotional relief	No	2.67 (0.76)	13.12 *	2.50 (0.64)	9.98 *	1.91 (0.67)	16.40 *
	Yes	3.09 (0.81)		2.78 (0.76)		2.39 (0.80)	

*d.f.* (2548). ^+^ *p* < 0.05; * *p* < 0.001.

**Table 3 ijerph-18-07653-t003:** Logistic regression models for thoughts and behaviors related to revenge.

Model	Likelihood Test Ratio χ^2^	Hosmer–Lemeshow, χ^2^	*p*	Predictors	β	*SE*	Wald’s χ^2^	OR	95% CI		Cohen’s *d*
									Lower Bound	Upper Bound	
Thinking about revenge	224.251 **	7.616	0.472	Machiavellianism	0.348	0.064	29.936 **	1.416	1.250	1.604	0.19
				Psychopathy	0.514	0.073	49.331 **	1.672	1.449	1.930	0.28
				Sex	0.365	0.088	17.165 **	1.441	1.212	1.712	0.20
				Infidelity	0.564	0.186	9.516 *	1.758	1.220	2.533	0.31
				Aggression	0.281	0.136	4.271 ^+^	1.324	1.015	1.728	0.15
				Jealousy and control	−0.409	0.120	11.531 **	0.664	0.525	0.841	−0.23
				Communication	−0.336	0.094	12.902 **	0.715	0.595	0.858	−0.19
Thinking about harming others	210.420 **	7.495	0.484	Psychopathy	0.914	0.074	152.770 **	2.495	2.158	2.884	0.50
				Aggression	−0.609	0.184	11.026 **	0.544	0.379	0.779	−0.34
				Jealousy and control	−0.888	0.196	20.515 **	0.411	0.280	0.604	−0.49
Attributing intentionality	177.106 **	7.191	0.516	Machiavellianism	0.273	0.064	18.227 **	1.314	1.159	1.490	0.16
				Psychopathy	0.406	0.072	31.931 **	1.502	1.304	1.729	0.22
				Narcissism	−0.214	0.076	7.843 *	0.807	0.695	0.938	−0.12
				Aggression	0.976	0.135	52.082 **	2.653	2.035	3.457	0.54
				Jealousy and control	−0.145	0.124	11.237 **	0.660	0.518	0.842	−0.08
				Communication	−0.271	0.096	7.954 *	0.763	0.632	0.921	−0.15
Dialogue to solve it	99.799 **	11.458	0.177	Psychopathy	–0.411	0.070	33.999 **	0.663	0.577	0.761	−0.23
				Infidelity	–0.585	0.249	5.511 ^+^	0.557	0.342	0.908	−0.32
				Jealousy and control	0.678	0.185	13.480 **	1.971	1.372	2.831	0.37
				Communication	0.394	0.141	7.846 *	1.483	1.126	1.954	0.22
Solution after dialogue	45.316 **	0.001	0.999	Infidelity	−0.794	0.186	18.264 **	0.452	0.314	0.651	−0.44
				Communication	0.338	0.093	13.204 **	1.401	1.168	1.681	0.19
Problem continues after dialogue	50.135 **	7.516	0.482	Psychopathy	0.220	0.055	16.040 **	1.246	1.119	1.387	0.12
				Sex	0.341	0.084	16.520 **	1.406	1.193	1.657	0.19
				Aggression	0.373	0.131	8.135 *	1.453	1.124	1.878	0.21
				Jealousy and control	0.281	0.118	5.710 ^+^	1.325	10.52	1.668	0.15
Resorting to police or justice	137.615 **	10.103	0.258	Machiavellianism	0.414	0.092	20.278 **	1.513	1.263	1.812	0.23
				Sex	0.440	0.157	7.857 *	1.553	1.142	2.113	0.24
				Infidelity	−1.042	0.320	10.582 **	0.353	0.188	0.661	−0.57
				Aggression	1.249	0.198	39.745 **	3.486	2.365	5.140	0.69
				Communication	−0.725	0.192	14.189 **	0.484	0.332	0.706	−0.40
Appraisal of revenge success	323.520 **	9.596	0.295	Machiavellianism	0.412	0.064	40.925 **	1.509	1.330	1.712	0.23
				Psychopathy	0.682	0.069	96.377 **	1.977	1.725	2.265	0.37
				Sex	0.405	0.093	18.970 *	1.499	1.249	1.799	0.25
				Infidelity	0.527	0.201	6.871 *	1.693	1.142	2.510	−0.28
				Jealousy and control	−0.396	0.133	8.906 *	0.673	0.519	0.873	−0.13
Assessment of not being discovered	324.770 **	9.225	0.324	Machiavellianism	0.508	0.071	51.335 **	1.662	1.446	1.910	0.28
				Psychopathy	0.676	0.074	83.569 **	1.966	1.701	2.273	0.37
				Sex	0.455	0.102	19.853 *	1.576	1.290	1.926	0.25
				Jealousy and control	−0.508	0.147	11.957 **	0.602	0.451	0.803	−0.28
				Communication	−0.238	0.113	4.429 ^+^	0.788	0.631	0.984	−0.13
Emotional relief	341.671 **	19.826	0.011	Machiavellianism	0.363	0.064	32.622 **	1.437	1.269	1.628	0.20
				Psychopathy	0.735	0.069	111.820 **	2.085	1.829	2.389	0.41
				Sex	0.444	0.092	23.065 *	1.588	1.300	1.867	0.24
				Infidelity	0.678	0.202	11.262 **	1.969	1.326	2.926	0.37
				Jealousy and control	−0.359	0.130	7.369 *	0.698	0.541	0.901	−0.20

Wald’s χ^2^ *d.f.* (1); ^+^ *p* < 0.05, * *p* < 0.01; ** *p* < 0.001.

## Data Availability

Data available at https://figshare.com/s/5f612d7e48bbda2c5a3e.

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
