# Peer review of "Revenge in Couple Relationships and Their Relation to the Dark Triad"

_ijerph, 2021, doi:10.3390/ijerph18147653_

Round 1

Reviewer 1 Report

Dear authors,

Congratulations on your work.

Best regards.

Author Response

Thank you for your comments and encouragement.

Reviewer 2 Report

Thank you for the opportunity to read your Manuscript titled “Revenge in couple relationships and their relation to the Dark Triad”. I find the article interesting, but I have a few rather some comments.

I suggest using keywords that are not included in the title to better facilitate searching.

The introduction is overall well written, but the study hypothesis and study motivation should be presented more in details.

I suggest separating the instruments with two separate sub-sections.

I consider that the authors should better explain how the participants were invited to participate in the study. 

How were the categories in the ad hoc questionnaire constructed? How many experimenters performed this categorization? It would be appropriate to explain this in the text.

In the data analysis section, it is necessary to write which statistical analysis software has been used and better specify the analysis carried out on the categories.

In the text, there are citations that do not fit the numerical model since the authors' names appear. Other formulations such as "A study conducted by university X" or "A study based on surveys Y" should be used, without mentioning the authors.

The value of the article could also be increased by extending the Discussion section with a discussion of the practical implications of the results.

Author Response

Thank you for the opportunity to read your Manuscript titled "Revenge in couple relationships and their relation to the Dark Triad". I find the article interesting, but I have a few rather some comments.

  • Thank you for your comments. They have really helped improving the manuscript.

I suggest using keywords that are not included in the title to better facilitate searching.

  • We believe this is a very good advice and now we have included as keywords: Psychopathy; Machiavellianism; dark personality; romantic relationships; retaliation.

The introduction is overall well written, but the study hypothesis and study motivation should be presented more in details.

  • The objectives and hypotheses of the study are now presented in more detail and are more sucinct (lines 102-116)

I suggest separating the instruments with two separate subsections.

  • Now we have different subsections for both measures in the study

I consider that the authors should better explain how the participants were invited to participate in the study.

  • Participant recruitment was made using a snow-ball technique with the aid of undergraduate collaborators. More information is now given in lines 135 to 140

How were the categories in the ad hoc questionnaire constructed? How many experimenters performed this categorization? lt would be appropriate to explain this in the text.

  • The categories were developed based on forensic practice by the authors. We give a brief description of how they were developed in lines 237 to 259

In the data analysis section, it is necessary to write which statistical analysis software has been used and better specify the analysis carried out on the categories.

  • We now provide a more detailed description of the initial analysis and state that SPSS 27 software was used in the analyses.

In the text, there are citations that do not fit the numerical model since the authors' names appear. Other formulations such as "A study conducted by university X'' or "A study based on surveys Y" should be used, without mentioning the authors.

  • Names of authors have been eliminated from the text and the manuscript has been redrafted accordingly.

The value of the article could also be increased by extending the Discussion section with a discussion of the practical Implications of the results.

  • In lines 501 to 512 we discuss how our results may be useful in intervention and in profiling high risk couple conflicts.

Round 2

Reviewer 2 Report

It's ok.

This manuscript is a resubmission of an earlier submission. The following is a list of the peer review reports and author responses from that submission.

Round 1

Reviewer 1 Report

Please find the attachment

Author Response

Dear authors, thank you for giving me the opportunity to review your manuscript: The components of revenge, intimate partner aggression and dark personality I think that this manuscript can meaningfully contribute to the literature. Given the relevance of your work, I suggest the following adjustments:

Line 44. Verify IPA-)

Line 48-50. The authors give a brief review of revenge. In these lines they present the aim of the study and again refer to revenge. I suggest they specify what objectives they describe in line 48-50, making it clear what these objectives refer to.

Line 51. I suggest deleting the question because it doesn't seem to add anything to the Introduction.

A: Thank you for these suggestions that improve the style of the manuscript. We have changed the text accordingly.

Line 56-58. The authors should provide a reference.

            A: Reference added

Some parts of the introduction appear in brief paragraphs, giving the impression that the authors are talking about different issues from the previous or the one that follows. I suggest integrating these ideas.

            A: Thank you for this advice. We have streamlined the text to make it clearer

Line 165. The authors should mention how they contacted the participants: 3942 participants were contacted

A: We had the assistance of students who participated in exchange for course credits and distributed a link among participants who subsequently responded to an online questionnaire. This is now stated in the text

Line 209. Verify “(5)).”

Line 213.

- Can you please include more information on the data collection (e.g., which social networks?).

- Were participants compensated for their time?

- Please indicate where the data was collected (country).

- There is considerable debate about the utility of on-line data. Can you please indicate in limitations section? The samples need to look at carefully.

Line 341. Typo “Jealousy”

Line 382 Typo “and and”

            A: Thank you, we have corrected these typos

Conclusion. I think the authors do not give themselves enough credit - there are several reasons this study has relevance. Why would it be important in practice? Please, can you show it in the manuscript?

Please expand on strengths of the study and underline the message they want to leave for the reader.

            A: Thanks.

Author Response

Dear authors,

thank you very much for giving me the opportunity to read your interesting

research idea with the amazing underlying sample size. Unfortunately, I have several major issues that bring me a little bit in trouble for an overall positive evaluation. Some can be easily addressed, but some are part of the study design. So, I really hope that you can shed light in my concerns or discuss limitations extensively.

Major issues

  1. Assessment of revenge process: as far as I understand correctly, you developed new items. However, the way in which the items were developed is rather unclear and the description is far behind from APA standards. That although revenge seems to be the main construct in your study. I am not versed in the area of revenge research. However, I had a quick view and I found further scales aside from the R scale like vengeance scale, transgression related interpersonal motivations inventory to name few. So, for me it is rather unclear why you developed new items. Moreover, for validation purposes you should have also assessed existing measures. Thus, for me you have to provide a strong rationale (based on theory and findings) why and HOW you developed new items. All items should be available for the readers.

            A: The questionnaire developed for this study measured specifically the revenge process resulting from a conflict in a couple relationship. Items were based on professional practice from the first author. We changed the text in the Instruments section to provide this information. All items were already included in the study.

Other existing questionnaires do not address specifically revenge within the couple. Since revenge is related to the Dark Triad, t scores showing significant differences in dark traits depending on positive or negative responses from participants support the validity of the questionnaire. However, not providing a comparison with other revenge questionnaires is a limitation that we address in the discussion.

  1. Another issue with regard to revenge is that it is not clear what kind of revenge is in your interest: clinical or subclinical? I guess subclinical, as the SD3 is also a subclinical measure. However, is the association to crime correct or relevant then? And would it be sufficient to focus on revenge within (intimate?) relationships? And why is your focus on break ups? What defines a traumatic break up? You are using a lot of strong terms that might not fit to what you are considering.

            A: Indeed, we assess subclinical revenge. When we include the term “crime” we mean it as the plausible ultimate result from revenge. However, we have altered instances were this an related terms appear to avoid conveying that crime is a inevitable consequence of revenge.

  1. On the other hand, with regard to your research question ‘association with the Dark Triad’

information depicted is restricted to 2 sentences ll 65-67.

            A: Indeed, information on the relationship between revenge and the dark triad was missing. Although there is a paucity of research on this issue, we have expanded this part of the introduction. We apologize for the previous omission.

  1. The title introduces three constructs, but you only use two measures.

            A: In the previous title, we used the term Intimate Partner Aggression (IPA) since revenge in couple relationships usually results in IPA. However, we agree that the title may have been misleading since we do not address actual IPA. We have changed the title accordingly.

  1. The research gap is also not clear to me. What is the information gain of your study?

            A: We provide additional data on the relationship between the Dark Triad and revenge in couple relationships, assess the frequency and characteristics of thoughts and actions related to research in couples. We hope to have made it clearer now with the new drafting of the text

  1. The introduction part seems to be quite unstructured to me and thus confusing. A strong

focus is lacking. You are using a lot of paragraphs that hinder to combine information depicted. I would delete ll31-36 as well as ll 72-86. Instead, I would focus on the relevant

aspect of revenge IN RELATION to the Dark Triad and aggression. I am not sure whether you submitted the manuscript to the Dark Triad special issue, but if this is the case, your

manuscript should focus on the Dark Triad, its relation to aggression and aggression theories with a focus on revenge. E.g., NA and PP might be more interested in the immediate revenge whereas MA might be more interested in long-term revenge which seems to be of your interest. This might also be the way to work out your research gap and give rationale to the revenge idea. It might also shift away the focus of revenge that does not seems to be assessed appropriate. By the way, you only cite 3 references that deal with the Dark Triad.

Even in case the focus on not on the DT this is not really well-established.

            A: We have redrafted the text so as to clarify the relationship between the Dark Triad and revenge in couples and provided more information on previous research in this issue.

  1. The statistical analyses part again is not APA conform. E.g., did you use RStudio? How about

requirements? Why did you choose the Dark Triad (not the dark triad) to be the first predictor and not gender? Why are the DT and type of conflict predictors? The type of conflict might be a moderator? Where is the aggression that you have mentioned in the title?

            A: We used SPSS 25 for the analyses. Requirements to participate in the study were aimed at providing a suitable sample to answer our research questions

            A: The Dark Triad and gender were both included in the analyses and both are significant predictors of revenge. We wanted to stress the role of dark traits in revenge. Our results show that psychopathy is often a stronger predictor than sex

            A: Following your suggestion we have changed every instance of the “dark triad” with “Dark Triad”.

  1. Group identification: what is the rationale for this kind of grouping? A clusteranalysis might be interesting for group identification or even for the entire research question (!).

            A: We divided participants in age groups merely for data collection and to get a wide and varied range of ages through stratified sampling. We didn’t use age groups in our analysis. Similarly, we initially divided participants according to whether they had felt themselves victims of some wrongdoing because the targets for our study were those participants who potentially felt entitled to revenge.

  1. The result part seems also quite unstructured to me. Where is your aim and the answer to it?

And findings should not be titled as interesting in the result part see l 258 which is by the way a descriptive result.

            A: We have made changes in the results section structure (i. e. table 3) and we hope to have improved its clarity.

            A: We also agree that is not appropriate to add epithets to results, and we have eliminated the label “interesting”. Thank you for pointing this out.

  1. Sample calculation is missing as well as effect sizes, CI, and power information. ‘Highly’ significant is no longer scientific standard see APA. A result is either significant or not (at a certain predefined level).

            A: As in the previous comment, we have eliminated the term “highly”.

            A: As a measure of effect size, we included odd ratios. However, since OR is a non-standarized measure of effect size, we now include Cohen’s d in table 3.

  1. If you choose to remain with regression analyes I would prefer to see the results in a table.

            A: We agree that results are better laid out in a table. For this purpose, we now made a table (table 3), which includes all the logistic regression results previously in the text.

  1. There seems to be a mixture of discussion and conclusions. That should be disentangled.

            A: Thank you for this suggestion. We hope to made these sections clearer now.

  1. The conclusion suffers from the issues mentioned above and below – likewise the abstract.

            A: We have made changes to these sections too.

minor issues

  1. As you can see my English could be improved, but I would suggest to use the same terms

consequently e.g., intimate partner vs. couple relationship. This is a little bit in line with what

I have written above. Focus on the constructs considered and do not open doors that you will

not be able to close / answer based on your data (e.g., crime, trauma).

            A: Thank you for your suggestion to avoid overstatements. We have changed the statements accordingly.

  1. the following sentences need references ll 28-30; 157-161

            A: The statements in those sentences have been modified to avoid controversial statements and now are more introductory general statements. As they stand now, we feel adding a reference is not appropriate.

  1. why did you use age groups? What is the rationale?

            A: As stated above, the sample was stratified to obtain an homogeneous number of participants sex and age wise.

  1. Again, there is a mixture of constructs traumatic event l 159 vs. conflict with partner l 167

(how was conflict defined)

            A: We have now unified terms across the text to avoid confusion. Conflict was defined based on the participants perception of a negative interaction event with their partner as defined in “2.1 participants”

  1. You probably had missing data?

            A: No, apart from participants that did not meet the criteria and were excluded from the study, (as specified in 2.1 participants) participants responded to an online questionnaire which prevented skipping answers.

  1. ll 214 which “test”?

            A: We meant measure. This term has now been changed.

  1. ll 214-216 repeating and contradicting information see above.

            A: This sentence did not really contribute to the text and we have suppressed it altogether

  1. Dark Triad, NA, MACH, PP not Mach Narc Psyc

            A: Thank you, we have also fixed these abbreviatures

  1. Exemplarily l 388 it should not be necessary to explain study design information in the result

Part.

            A: We agree that repeating the procedure is not needed so we have changed the phrasing of this paragraph

Round 2

Reviewer 2 Report

Dear authors,

thank you very much for your responses and revisions. They helped to clarify some issues. However, I apologize for still being a bad cop. Please see my concerns below.

major

  1. I am still not convinced of your title: you developed a scale but write “the components…” Aside, ‘dark personality’ might address further traits like sadism.
  2. I think the sentence in the ‘background part ‘ still does not fit to your research question and again not to the title.
    1. If you don’t use age groups in your analyses, why is it relevant to your research question then / must be mentioned?
    2. Participants filled out a self-developed questionnaire with regard to revenge.
    3. Is Jones & Paulhus a relevant information in the abstract?
    4. The first sentence of the result part is hard to understand and does not include any statistical information.
    5. If you write independently from the other dark traits this is not correct; you probably mean when controlling for MACH and PP. Moreover, in one sentence your write that NA is related but in the following sentence it is not.
    6. I rest at my case the crime and gender-based violence is not part of you study. I think you still fail to focus on what your study adds to knowledge and on what sidelines are.
  3. Intimate partner aggression is still part of the key words.
  4. L 29. the cited reference does not fit to the content of the sentence. The sentence still does not emphasize what the research gap is. Again, in the following sentence references are missing.
  5. I rest at my case that you should have a stronger focus on research instead of story telling (e.g., ll33-38). This information is followed by one reference from 2016.
  6. I am confused about the citation 3: I have only checked the abstract but they write about intimate partner aggression but you use this reference to explain revenge. I am sorry, but in my view, you still mix up mix up terms and constructs. And I still have the impression that the manuscript is kind of knitted quickly so I will stop going in detail in the following as my review will probably fill several pages.
  7. Further examples are the new added sentence ll 109: rationale/reference is missing.
  8. I appreciate the added information with regard to the DT ll 63-73 but where is the relation to the research question?
  9. L76 why is the children information relevant to your research question?
  10. Your description of the new developed scale still does not fit to APA criteria.
  11. I still rest at my suggestions for statistical analyses – also with regard to APA standards.
  12. Ll 384 why is it especially relevant in the case of infidelity?
  13. As far as I know IJERPH needs a discussion and a conclusion part.

In sum, even though you addressed some of my issues successfully, the manuscript still suffers from several flaws. This also because the revisions were not carried through consistently. I therefore suggest to reject the manuscript in the present form, but I would recommend a new submission.